# 8–17 DNAzyme Silencing Gene Expression in Cells via Cleavage and Antisense

**DOI:** 10.3390/molecules28010286

**Published:** 2022-12-29

**Authors:** Zhongchun Zhou, Wen Sun, Zhen Huang

**Affiliations:** 1Key Laboratory of Bio-Resource and Eco-Environment of Ministry of Education, College of Life Sciences, Sichuan University, Chengdu 610064, China; 2SeNA Research Institute & Szostak-CDHT Large Nucleic Acids Institute, Chengdu 610041, China

**Keywords:** 8–17 DNAzyme, gene silencing, catalysis, antisense, chemical modifications

## Abstract

Gene silencing is an important biological strategy for studying gene functions, exploring disease mechanisms and developing therapeutics. 8–17 DNAzyme is of great potential for gene silencing, due to its higher RNA-cleaving activity. However, it is not generally used in practice, due to its divalent cation dependence and poor understanding of its cellular mechanisms. To address these issues, we have explored its activity in vitro and in cells and found that it can cleave RNA substrates under the simulated physiological conditions, and its gene-silencing activity is additionally enhanced by its RNase H compatibility, offering both cleavage and antisense activities in cells. Further, chemical modifications can facilitate its stability, substrate binding affinity and gene-silencing activity. Our research results suggest that this DNAzyme can demonstrate high levels of activities for both actions in cells, making it a useful tool for exploring biomedical applications.

## 1. Introduction

Gene silencing takes place naturally in cells by suppressing the expression of a gene at transcriptional or translational levels [1]. Following the discovery and demonstration of RNA interference (RNAi) [2], many astounding advances have been accomplished through gene silencing, in studying gene functions [3], inter-genic interactions and molecular mechanisms of diseases. Most importantly, it is a promising strategy to tackle various diseases, as many diseases are caused by the wrong expression of genes.

There are many gene-silencing methodologies based on nucleic acid sequences, such as antisense oligonucleotides (ASOs), small interfering RNAs (siRNAs), microRNAs (miRNAs), ribozymes, DNAzymes and others [4,5], making it possible to treat any difficult diseases. Among them, ASOs and siRNAs have been approvedfor use as therapeutics in disease treatments [6]. However, they still have shortcomings, for example, siRNAs are of relatively poor stability, immunogenicity and “off-target” effects in vivo, and ASOs have relatively low efficacy and high “off-target” effects when used at high concentrations in practices [7]. Although both RNA-cleaving ribozymes and DNAzymes can specifically cleave RNA substrates, DNAzymes are simpler and more stable than ribozymes. Further, DNAzymes have many advantages, including stability, small molecular size, high selectivity, designability, modifiability and availability [8], making them potential gene-silencing tools.

Two chiefly studied RNA-cleaving DNAzymes are 10–23 DNAzyme (Dz10–23) and 8–17 DNAzyme (Dz8–17), due to their RNA-cleaving activities and substrate selection in vitro and in cells [9]. Each consists of a catalytic loop (13–15 nt) and two substrate-binding arms of variable lengths and sequences, recognizing RNA substrates through the Watson–Crick base pairing [10]. While Dz10–23 is successfully used for gene silencing [11,12], it cannot be applicable to every target gene due to its cleavage site choice and its accessibility to RNA targets. 8–17 DNAzyme, with the determined structure [10] for rational design, is presumably of greater potentials for gene silencing. Unfortunately, research on Dz8–17 is still not sufficient for silencing gene expression efficiently in cells. The lower utility of Dz8–17 is attributed to the poor understanding of its cellular mechanisms and divalent cation dependence [10,13]. Since the physiological ionic strengths of free Mg^2+^ (approximately 0.5–1.0 mM) [14] are reasonably high, likely sufficient to support DNAzyme catalysis, other reasons (such as the DNAzyme stability, nonspecific binding, and target accessibility) might play major roles in causing its poor cellular activities.

To face these problems, we decided to explore the in vitro cleavage of Dz8–17 under simulated physiological conditions, Dz8–17 modifications, and target accessibility. DNAzymes without any modifications can be quickly cleaved by endonucleases and exonucleases in cells and serum. Therefore, in order to make DNAzymes more active in cells, molecular modifications are necessary. Chemical modification is currently a smart strategy to maintain the intracellular activity of DNAzymes. Most commonly, phosphorothioate (PS) linkages are used. PS modification was first used to protect ASOs from the degradation of serum exonuclease and endonuclease [15], and later it was also used to protect DNAzymes [16]. Compared with unmodified DNAzymes, PS-modified DNAzymes have reduced catalytic activity, but their ability to inhibit gene expression in cultured cells is much stronger than that of their unmodified form [17], as it is more resistant to cellular nuclease degradation. The second most popular is 2′-O-Me (OMe), a natural modification widely used for ASOs and aptamers [18]; it can also protect DNAzymes from nuclease degradation [19]. The third most popular is locked nucleic acids (LNAs), a remarkable duplex stabilizer, which has been successfully applied to the modifications of siRNAs [20] and ASOs [21]. Although chemical modifications can enhance the gene-silencing activity of Dz10–23 in cells [16,19], the intracellular activity of chemically modified Dz8–17 remains unknown.

Further, we investigated its cellular activity using the mRNA of EGFP as the target, which is broadly used in gene-silencing studies [12,22]. For identifying the accessible target mRNA sites in cells, we designed a set of Dz8–17 enzymes and examined their activities on cleaving the mRNA fragment in vitro. Interestingly, we found that some Dz8–17 enzymes identified in vitro also worked best in cells, indicating the consistency between in vitro and cellular activities. In addition, for the activity consideration, we studied this DNAzyme and performed in vitro and cellular investigations in order to provide insights into the mechanisms and cellular activities of Dz8–17. Finally, we successfully identified the chemically modified Dz8–17 with both cleavage and antisense activities on EGFP mRNA in cells (Figure 1).

## 2. Results and Discussion

### 2.1. The Design of DNAzymes Targeting EGFP mRNA

According to the EGFP mRNA sequence (Appendix A) and cleavage site of Dz8–17 (AG or GG) and the basis of bioinformatic analysis, we designed and synthesized a set of Dz8–17 DNAzymes (12 DNAzymes, Dz01–Dz12, Table 1) targeting the EGFP-coding region.

### 2.2. Identification of the DNAzyme with Gene-Silencing Activity in Cells

First, we transcribed the full-length mRNA of EGFP (882 nt, Appendix A) as the long substrate for these twelve DNAzymes. In order to better investigate the cleavage of Dz8–17 in cells, we simulated physiological conditions for performing in vitro cleavage experiments. The in vitro cleavage studies indicated that eleven of the twelve Dz8–17 enzymes had various degrees of cleavage activity (Figure 2a). We found that Dz04 and Dz08 had relatively higher activities (Figure 2a). By using short RNA substrates (23 nt, Appendix A), we confirmed the catalytic activities of Dz04 and Dz08 (data not shown).

Later, we tested the eleven DNAzymes for gene-silencing activities in cells. To resist nuclease degradation, two PS linkages were introduced to each end of their binding arms. Then, they were co-transfected into 293T cells with EGFP expression plasmids (transient transfection). By measuring the intensity of EGFP fluorescence after 48 h, we found that Dz04 had the strongest activity in gene silencing (Figure 2b, Table 2). We performed a secondary structure prediction of mRNA using RNAfold, and found that the target region (206–228) of Dz04 formed a big open window, which can ease the target binding for Dz04. Although some Dz8–17 enzymes such as Dz06 and Dz08 had relatively higher cleavage activities in vitro, they had poor gene-silencing effects due to their poor target accessibility in cells.

To confirm the substrate-specific recognition of DNAzyme Dz04, we incorporated two mismatches into each arm individually or both (Appendix A). We found that after incorporating two mismatches on each arm (Figure 3a), the silencing activity was reduced (compared with the wildtype Dz04), while it was most reduced when there were four mismatches on both arms. Interestingly, the two mismatches on the left arm caused a stronger reduction in activity than those on the right arm.

Further, to confirm the activity–concentration relationship, we investigated various DNAzyme concentrations in the cellular experiments. The silencing activity was concentration-dependent (Figure 3b). These results collectively indicated that Dz04 indeed targeted EGFP mRNA, causing cellular gene silencing. Our results suggest that the physiological ionic strengths may be sufficient for DNAzyme catalysis in cells. In addition, the in vitro catalytic experiments can guide efficient DNAzyme identification in cells.

### 2.3. The DNAzyme with Both Cleavage and Antisense Activities in Cells

Our experiments above indicated that DNAzyme Dz04 is fairly active in vitro and in cells. To clarify its cellular mechanisms, we mutagenized the DNAzyme by changing T_1_ to C_1_ (Dz4M, Figure 4a,b), inactivating its cleavage activity without perturbing the overall structure (Figure 4c). Dz4M can bind to its target through hybridization as well as Dz04 (Figure 4b). We co-transfected PS-modified Dz04 and Dz4M (with two PS linkages at both 5′ and 3′ ends, Appendix A) individually into 293T cells with EGFP expression plasmids. Twenty-four hours later, after transfection, we quantified the RNA expression levels by RT-PCR. We found that compared with the mock control, Dz04 offered a 70% reduction in gene silencing, while catalytically inactive Dz4M merely offered a 50% reduction (Figure 4d and Appendix A). This difference (20%, *p* < 0.05) indicated that Dz04 was catalytically active in cells. In addition, the cellular activity with this single-base mutation suggested that Dz04 had antisense activity as well. Our results confirmed our hypothesis that the physiological ionic strengths are sufficient for DNAzyme catalysis in cells. However, the antisense nucleic acid effect was dominant in the gene-silencing activity of Dz8–17, which is consistent with the 10–23 DNAzyme [23,24].

### 2.4. In Vitro Studies of the Stability, Catalysis and RNase H Compatibility of the DNAzymes Modified on the Arms

To increase the biological stability of the DNAzyme, we introduced modifications at its ends OMe and LNA, in addition to the PS linkages, to increase duplex binding and reduce degradation by intracellular nucleases [11]. However, on the basis of our investigations above, we need to balance chemical modifications with the stability, DNAzyme cleavage activity and antisense function (such as RNase H compatibility or RNase H-caused RNA digestion).

To address these issues collectively, we designed and synthesized the DNAzyme with PS, OMe and LNA modifications on the arms (Figure 5a). First, we examined the stability of these modified enzymes (Appendix A) in fetal bovine serum (FBS). We found that LNA was the most stable and PS and OMe were less stable, while non-modified Dz04 was almost completely degraded by FBS within 24 h (Figure 5b).

Second, we performed the in vitro cleavage of the modified Dz04 on the short RNA substrate (Appendix A) and we found that the chemical modifications did not alter their activity significantly, excluding Dz04-4Ome and Dz04-4s(4Ome)m (Figure 5c). Third, to investigate the impacts of the modifications on the antisense function (such as RNase H compatibility), we performed in vitro experiments on RNase H-directed hydrolysis of RNA. The results showed that the OMe and LNA in the middle of each arm reduced the RNase H compatibility (Figure 5d), while the PS, OMe and LNA modifications at both ends did not.

Our results indicated that most of these modifications do not significantly reduce the cleavage activity of Dz8–17, and the two modifications (such as LNA) at each end are sufficient for the stability requirements. Since OMe and LNA in the middle of the arms are incompatible with RNase H-mediated RNA degradation [25], OMe and LNA at the end of each arm allowed for RNase H compatibility (antisense-mediated cleaving activity).

### 2.5. The Modified DNAzyme with High Activities in Cells

To find out the optimal number of modified nucleotides on each end, we designed Dz04 containing one, two, three and four PS linkages at both ends (Appendix A). Our results showed that the modified DNAzymes had higher activity than the unmodified one, while the higher number of PS did not significantly increase the cellular activity (Figure 6a), indicating that two modified nucleotides at each end (for PS linkages) were sufficient.

To further explore the gene-silencing activities of DNAzymes with different chemical modifications, we transfected cells with DNAzymes containing the PS, OMe or LNA modifications with EGFP-expression plasmids, and analyzed the fluorescence intensity after 48 h. We discovered that at the DNAzyme concentration of 100 nM, the gene-silencing efficiencies of DNAzymes with terminal LNA, PS and OMe (Figure 6b,c; Appendix A) were 82, 60 and 36%, respectively. Further, we found that the LNA-modified cleaving DNAzyme (Dz04-4LNA) offered 82% inhibition, while the mutant (Dz4M-4LNA) without cleaving activity offered 75% inhibition. In addition, OMe or LNA combined with the terminal PS did not further increase the activity. Moreover, the introduction of OMe and LNA in the middle of the arms largely reduced the cellular activities of DNAzymes, consistent with previous in vitro studies.

In order to eliminate the possibility of non-specific gene silencing by the chemical modifications, we designed scrambled controls with the same modifications (Dz4S, the scrambled sequences of the corresponding Dz04, Appendix A). As expected, we found that none of the scrambled controls had gene-silencing activities (Figure 6b,c).

Since LNA modifications are incompatible with RNase H-mediated RNA degradation [25], we have concluded that in cells, DNAzymes modified with two LNAs at each end have higher stability, catalytic activity, RNA substrate affinity (antisense-mediated arresting activity) and RNase H compatibility (antisense-mediated cleaving activity), thereby offering fine gene-silencing tools.

## 3. Materials and Methods

### 3.1. Materials

293T cell lines and pEGFP-C1 plasmids were derived from our laboratory. Growth media (DMEM) was supplemented with 10% FBS and 1% antibiotic–antimycotic. 293T cell lines were maintained up to 30 passages in a 37 °C/5% CO_2_ incubator. All oligonucleotides were synthesized by Sangon (Shanghai, China). Trypsin EDTA and penicillin streptomycin were purchased from Biosharp (Hefei, China). DMEM basic media and fetal bovine serum and Opti-MEM were purchased from Gibco. Lipofectamine 3000 was purchased from Invitrogen. A CCK-8 kit was purchased from AbMole. A HiScribe™ T7 in vitro Transcription Kit and RNase H were purchased from NEB. A Total RNA Isolation Kit, HiScript III RT SuperMix for qPCR, and ChamQ Universal SYBR qPCR Master Mix were purchased from Vazyme (Nanjing, China).

### 3.2. Methods

#### 3.2.1. Bioinformatics Tools for Designing 8–17 DNAzymes

We used OLIGO Primer Analysis Software (Oligo 7) to design DNAzymes semi-automatically (the oligo length and the Tm value were set at 23 nt and 65–80 °C, respectively). The secondary structures of DNAzymes were analyzed by Predict a Bimolecular Secondary Structure Web Server (https://rna.urmc.rochester.edu/RNAstructureWeb (accessed on 9 September 2021)) “URL”. The binding specificities of designed DNAzymes were analyzed by the basic local alignment search tool (BLAST) from NCBI (www.ncbi.nlm.nih.gov/BLAST (accessed on 10 September 2021)) “URL” to avoid binding to the host transcriptome non-specifically.

#### 3.2.2. In Vitro Substrate Cleavage

For the long RNA substrates (882 nt), the reaction components were as follows: 5 μL 2× Reaction buffer (100 mM HEPSE, pH 7.5, 300 mM K^+^, 1 mM Mg^2+^), 1 μL RNA substrates (300 ng/μL), 1 μL DNAzymes (10 μM), 3 μL DEPC ddH_2_O. For short RNA substrates (23 nt), the reaction components were as follows: 5 μL 2× Reaction buffer, 1 μL RNA substrates (2 μM), 1 μL DNAzymes (1 μM), 3 μL DEPC ddH_2_O. The above reaction components were mixed thoroughly and then incubated at 37 °C for 2 h. After the reaction, the cleavage products were analyzed by agarose gel electrophoresis (882 nt) or PAGE (23 nt).

#### 3.2.3. Co-Transfection of 293T Cells with DNAzymes and pEGFP-C1

293T cells were plated in a 96-well plate (2 × 10^4^ cells/well, four replicates/sample) or a 12-well plate (2 × 10^5^ cells/well) the day before co-transfection, and then were co-transfected with 100 ng pEGFP along with different DNAzymes (final concentration of 100 nM or 400 nM) using Lipofectamine 3000 according to the manual’s instructions. After 48 h of incubation, the cells expressing EGFP were scanned for fluorescence intensity, measured with a microplate reader (Varioskan LUX, Thermofisher, Beijing, China) or visualized with fluorescence microscopy (Nikon, Beijing, China).

#### 3.2.4. RNA Quantification (qRT-PCR)

Twenty-four hours after transfection, RNA was extracted from the transfected cells using a FastPure Cell/Tissue Total RNA Isolation Kit according to the protocol. HiScript III RT SuperMix for qPCR was applied for genomic DNA removal and cDNA synthesis. Real-time PCR reactions were carried out in LightCycler^®^ 96 (Roche, Shanghai, China) using ChamQ Universal SYBR qPCR Master Mix according to the protocol. The expression of EGFP gene was normalized to the GAPDH, and the primers are listed as follows: EGFP forward primer: 5′-CGGCAAGCTGACCCTGAA-3′, EGFP reverse primer: 5′-GACGTAGCCTTCGGGCA-3′; GAPDH forward primer: 5′-ACCCACTCCTCCACCTTTGA-3′, GAPDH reverse primer: 5′-CTGTTGCTGTAGCCAAATTCGT-3′.

#### 3.2.5. Stability Tests of Modified DNAzymes in FBS

The reaction components were assembled as follows: 1.0 μL DNAzymes (10 μM), 0.5 μL FBS and 8.5 μL ultrapure water. The reaction mixes were incubated at 37 °C for 0 h, 6 h, 12 h, 24 h or 48 h. Reactions were stopped by addition of 10 μL denaturing gel-loading buffer and denatured at 95 °C for 2 min. Finally, 5 μL denatured products were placed on a 16% PAGE denaturing gel for electrophoresis and analysis.

#### 3.2.6. RNase H-Induced Hydrolysis of RNA

The reaction components were assembled as follows: 5.0 μL 2× RNase H Reaction buffer, 1.0 μL RNA substrates (2.0 μM), 1.0 μL DNAzymes, 0.1 μL RNase H, 2.9 μL DEPC H_2_O. The reaction mixes were incubated at 37 °C for 10 min. Finally, the products were placed on a 16% PAGE denaturing gel for electrophoresis and analysis.

#### 3.2.7. Statistical Analysis

Student’s *t*-tests were used in GraphPad Prism 5.0 to analyze variances between groups (* *p* < 0.05, ** *p* < 0.01, *** *p* < 0.001 and **** *p* < 0.0001 represented significant results).

## 4. Conclusions

In summary, our research results provided insights into the catalytic activities and cellular antisense mechanisms of Dz8–17. In the presence of minimal divalent cations, it is catalytically active, and its gene-silencing activity is additionally enhanced by its RNase H compatibility in cells. In addition, we found that chemical modifications (such as LNA) can greatly enhance its activity in cells. Our investigations can also provide guidance for designing Dz8–17, such as in optimizing the arm lengths and placing modifications at the terminal positions to take advantage of the RNase H compatibility. Our studies demonstrated that under physiological conditions, the modified 8–17 DNAzyme can demonstrate high activities collectively regarding catalysis and antisense, making it a useful tool for exploring genetic functions and biomedical applications.

## Figures and Tables

**Figure 1 molecules-28-00286-f001:**
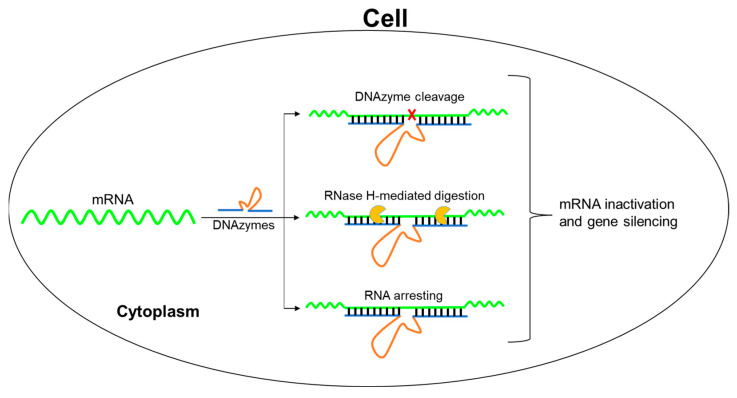
8–17 DNAzyme exerts high gene-silencing effect via catalytic cleavage and antisense activities.

**Figure 2 molecules-28-00286-f002:**
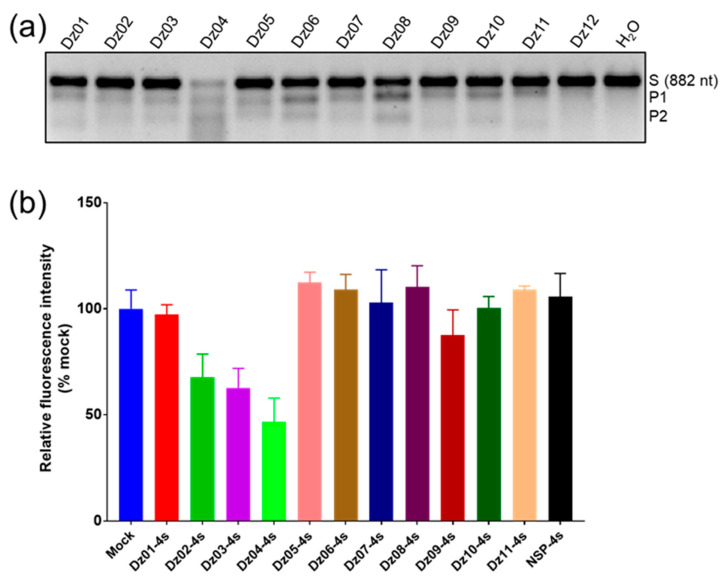
Finding the DNAzyme with gene-silencing activity in cells. (**a**) In vitro cleavage study of twelve Dz8–17 DNAzymes via agarose gel analysis. All reactions were carried out under simulated physiological conditions (150 mM KCl, 0.5 mM MgCl_2_, pH 7.5) at 37 °C for 2 h. s: substrate; P1: product 1; P2: product 2. (**b**) EGFP gene-silencing study with the DNAzymes in cells. The fluorescent data were collected 48 h after transfection and reported as mean value ± SD of three repeats. The transfection concentrations of the DNAzymes were 400 nM, and Mock was a negative control using ultrapure water instead of a DNAzyme. NSP: negative DNAzyme control with nonspecific arm sequences. Each DNAzyme and control were labeled with different colors.

**Figure 3 molecules-28-00286-f003:**
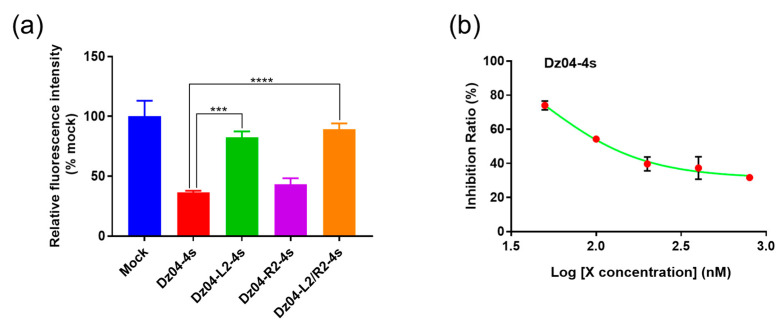
Identification the DNAzyme (Dz04) gene-silencing activity in cells. (**a**) The mismatched arm sequences of Dz04 compromised the EGFP gene silencing, and the data significance was determined by a Student’s *t*-test (*** *p* < 0.001, **** *p* < 0.0001); Each DNAzyme and mock control were labeled with different colors. (**b**) The gene-silencing activities of Dz04 at different concentrations.

**Figure 4 molecules-28-00286-f004:**
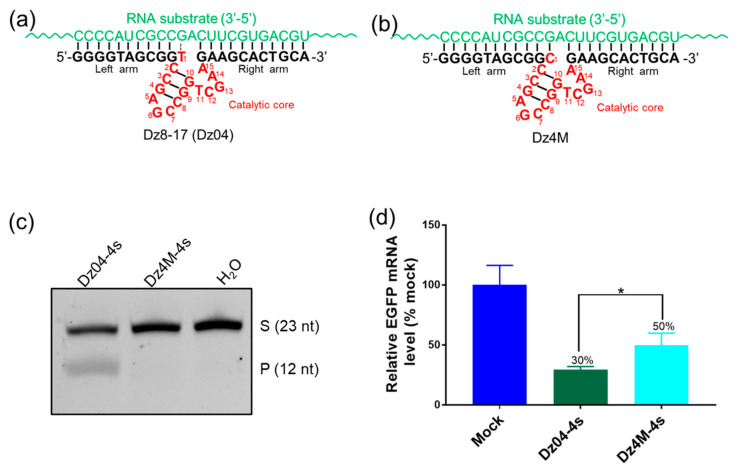
The DNAzyme (Dz04) with both cleavage and antisense activities in cells. (**a**,**b**) are secondary structures of Dz04, Dz4M and Dz04-4s. (**c**) The in vitro cleavage activities of Dz04-4s and Dz4M-4s under simulated physiological conditions. All reactions carried out in 1.0 μM Dz8–17, 1.0 μM substrates, 150 mM KCl and 0.5 mM MgCl_2_ at 37 °C for 2 h. (**d**) The cellular mRNA degradation by Dz04-4s and Dz4M-4s was analyzed by RT-qPCR, 24 h after the DNAzyme transfection. Stu-dent’s *t*-tests were used to assess significance of the data according to mean values and SDs of three repeats (* *p* < 0.05). Dz04, Dz4M and mock control were labeled with different colors.

**Figure 5 molecules-28-00286-f005:**
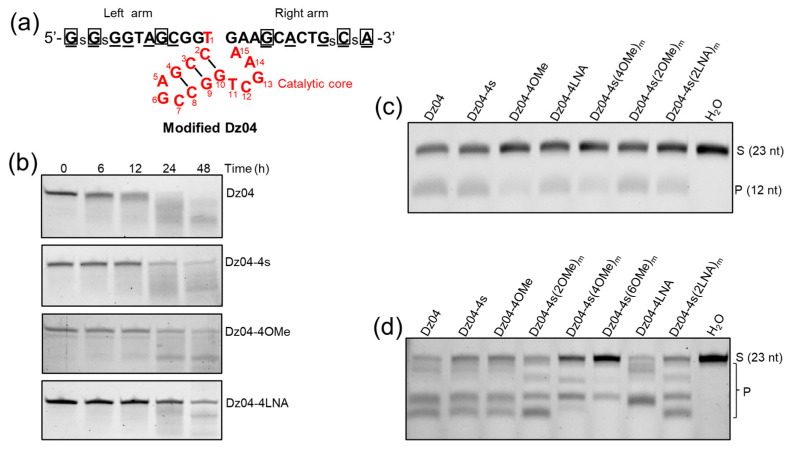
In vitro studies of stability, catalysis and RNase H compatibility of the DNAzymes with arm modifications. (**a**) Phosphorothioate linkage (PS), 2′-O-methyl ribonucleotide (OMe), 2′-O and 4′-C LNA (LNA) are represented by “s”, “underlined” and “box” labels, respectively. (**b**) Stability study of the modified DNAzymes under FBS serum treatment. (**c**) In vitro cleavage study of the modified DNAzymes under physiological conditions. S: substrate; P: product. (**d**) Modified DNAzyme compatibility with RNase H.

**Figure 6 molecules-28-00286-f006:**
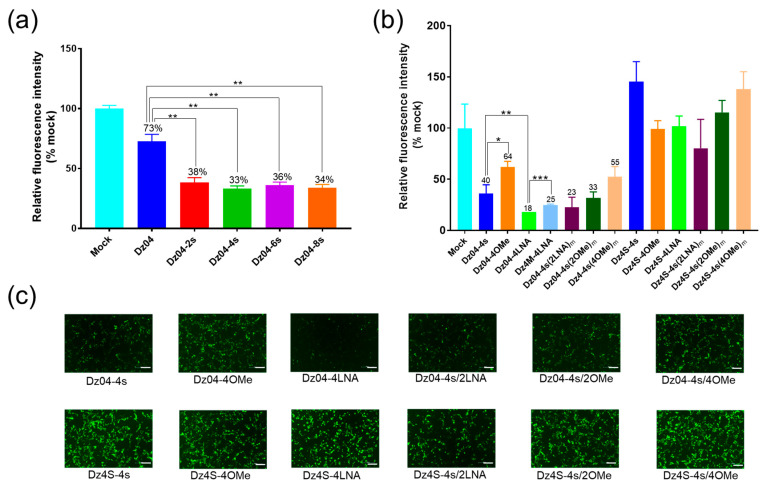
The modified DNAzyme (Dz04) with high gene-silencing effect in cells. (**a**) Effect of Dz04 PS modifications on gene silencing. Statistical significance was determined by a Student’s *t*-test of three repeats (** *p* < 0.01). (**b**) Gene silencing of Dz04 with different chemical modifications. The fluorescent data significance was determined by a Student’s *t*-test (* *p* < 0.05, ** *p* < 0.01, *** *p* < 0.001). Dz04-4s, Dz04-4OMe and Dz04-4LNA contained two PS, OMe and LNA modifications at both 5′ and 3′ ends of Dz04, respectively. Dz04-4s(2LNA)m contained one LNA in the middle of each Dz04-4s arm. Dz04-4s(2OMe)m and Dz04-4s(4OMe)m contained one and two OMe modifications in the middle of each Dz04-4s arm, respectively; Dz4M-4LNA contained two LNAs at both 5′ and 3′ ends of Dz4M (Figure 4b). Dz4S-4s, Dz4S-4OMe and Dz4S-4LNA contained two PS, OMe and LNA modifications at both 5′ and 3′ ends of Dz4S, respectively. Dz4S-4s(2LNA)m contained one LNA in the middle of each Dz4S-4s arm. Dz4S-4s(2OMe)mand Dz4S-4s(4OMe)m contained one and two OMe modifications in the middle of each Dz4S-4s arm, respectively. Their detailed sequences are shown in Appendix A. (**c**) Cellular fluorescent images of EGFP gene silencing with the modified DNAzymes in Figure 6b. Scale bar: 500 μm.

**Table 1 molecules-28-00286-t001:** The sequences of twelve Dz8–17 enzymes targeting EGFP mRNA.

Name	Sequence (5′-3′) ^1^	Target Region	Tm (°C)
Dz01	GGGCAGCTTGTCCGAGCCGGTCGAAGGTGGTGCAGA	143–165	72.5
Dz02	GCACTGCACGTCCGAGCCGGTCGAAGTAGGTCAGGG	191–213	70.0
Dz03	GGCTGAAGCATCCGAGCCGGTCGAAGCACGCCGTAG	198–220	70.8
Dz04	GGGGTAGCGGTCCGAGCCGGTCGAAGAAGCATGCA	206–228	70.1
Dz05	AGAAGTCGTGTCCGAGCCGGTCGAAGCTTCATGTGG	231–253	65.6
Dz06	GACGTTGTGGTCCGAGCCGGTCGAAGTTGTAGTTGT	431–453	63.7
Dz07	CGTTCTTCTGTCCGAGCCGGTCGAATGTCGGCCATG	459–481	66.5
Dz08	TGCCGTTCTTTCCGAGCCGGTCGAAGCTTGTCGGCC	462–484	70.9
Dz09	TGCAGATGAATCCGAGCCGGTCGAATCAGGGTCAGC	126–148	65.5
Dz10	TGAAGTTCACTCCGAGCCGGTCGAATGATGCCGTTC	477–499	63.8
Dz11	GAGCTGCACGTCCGAGCCGGTCGAAGCCGTCCTCGA	515–537	73.7
Dz12	ACGCTGCCGTTCCGAGCCGGTCGAATCGATGTTGTG	508–530	68.8

^1^ Red font, catalytic core sequence of Dz8–17.

**Table 2 molecules-28-00286-t002:** The inhibition ratios of 11 DNAzymes targeting EGFP mRNA and NSP control.

DNAzymes	Inhibition Ratio (%)	DNAzymes	Inhibition Ratio (%)
Dz01	<5	Dz07	0
Dz02	32	Dz08	0
Dz03	37	Dz09	12
Dz04	53	Dz10	0
Dz05	0	Dz11	0
Dz06	0	NSP	0

## Data Availability

Not applicable.

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
