# Peer review of "8–17 DNAzyme Silencing Gene Expression in Cells via Cleavage and Antisense"

_molecules, 2022, doi:10.3390/molecules28010286_

Round 1
Reviewer 1 Report
In this work the authors studied the 8-17 DNAzyme for the silencing of an mRNA. The study was performed inside cells and using inactive DNAzymes, the authors concluded that there are both cleavage and antisense effects, which is reasonable. I think it can be published after addressing the following concerns.
1. Table 1, would be nice to indicate the region of the mRNA each DNAzyme is targeting (e.g. from xxx to xxx).
2. Table 2 should have error associated with the numbers.
3. Can you comment on the comparison of 10-23 and 8-17 under physiological conditions. You don’t need to do the experiments on 10-23 but there are a lot of literature information out there for discussion.
4. What’s the cleavage rate of your 8-17 under the condition of your experiment? I think this is an important number to have and it will be useful to compare with previous and future work.
Author Response
Response to Reviewer 1 Comments
Point 1: Table 1, would be nice to indicate the region of the mRNA each DNAzyme is targeting (e.g. from xxx to xxx)..
Response 1: As suggested, the region of the mRNA each DNAzyme is targeting is indicated in Table 1.
Point 2: Table 2 should have error associated with the numbers.
Response 2: Some errors associated with the numbers has been corrected in Table 2.
Point 3: Can you comment on the comparison of 10-23 and 8-17 under physiological conditions. You don’t need to do the experiments on 10-23 but there are a lot of literature information out there for discussion.
Response 3: Both 10-23 and 8-17 DNAzymes cleave all-RNA substrates with Kcat ≈ 0.01 min−1 under simulated physiological conditions (2mM Mg2+)( Proc Natl Acad Sci. 1997; 94(9): 4262–4266). Noteworthily, first- and second-order rate constants fall 2–4 orders of magnitude under simulated physiological conditions (0.5 mM Mg2+) (Chem. Sci., 2018,9, 1813-1821). One study showed that at 0.5 mM Mg2+, Dz10-23 collapses into a catalytically inactive conformation (J Bio Chem; 2003; 278 (48): 47987-96.). Under higher Mg2+ concentration, 8-17 may exert higher cleavage activity than 10-23 in vitro (0.220 min-1 vs 0.103 min-1) (Mol. BioSyst., 2015, 11, 1454).
Point 4: What’s the cleavage rate of your 8-17 under the condition of your experiment? I think this is an important number to have and it will be useful to compare with previous and future work.
Response 4: We just have measured the cleavage rate of 8-17 (Dz04) in our another work, and its Kobs was 0.071 min-1 (under 10 mM Mg2+).
Reviewer 2 Report
In this manuscript, Zhongchun Zhou et al. investigated RNA-cleaving activity of 8-17 DNAzyme in vitro and in cell, in detail. The authors found that Dz04 sequence can effectively cleave a substrate mRNA encoding EGFP, both in vitro and in cell. Effect of PS, OMe and LNA modifications on cleavage activity and stability was also investigated, and the authors concluded DNAzyme modified with two LNAs at each end was the best design.
The manuscript has been well written, so I agree with publication after minor revision.
I am wondered why Dz04 showed remarkably high activity among Dz01-Dz12. Is this preference caused by secondary-structure of mRNA, which derives difference in accessibility? If so, prediction of secondary-structure of mRNA, using RNAfold or other tools, should be performed and discussed. If it is just caused by difference in cleavage activity depending on sequence, please provide clear discussion about the mechanism.
Author Response
Response to Reviewer 2 Comments
Point 1: I am wondered why Dz04 showed remarkably high activity among Dz01-Dz12. Is this preference caused by secondary-structure of mRNA, which derives difference in accessibility? If so, prediction of secondary-structure of mRNA, using RNAfold or other tools, should be performed and discussed. If it is just caused by difference in cleavage activity depending on sequence, please provide clear discussion about the mechanism..
Response 1: We have performed the prediction of secondary-structure of mRNA using RNAfold, and found that the target region (206-228) of Dz04 formed a big open window (Fig 1), which can explain why Dz04 showed remarkably high activity among Dz01-Dz12.
Figure 1. Local secondary-structure of mRNA of target region of Dz04.
